# Impact of electronic palliative care coordination systems (EPaCCS) on care at the end of life across multiple care sectors, in one clinical commissioning group area, in England: a realist evaluation protocol

Lucy Pocock,[1] Lydia French ,[1] Michelle Farr,[2] Richard Morris,[1] Sarah Purdy[1]

¹Centre for Academic Primary Care, University of Bristol, Bristol, UK
²NIHR ARC West, Bristol, UK

**Correspondence to**
Dr Lucy Pocock;
lucy.pocock@bristol.ac.uk

## ABSTRACT

**Introduction** Electronic palliative care coordination systems (EPaCCS) aim to support people approaching the end of life (EOL) to receive consistent care, according to their wishes, that is coordinated effectively across multiple care sectors. They are in use across the UK although empirical evidence into their effectiveness is poor. This paper presents a protocol of a mixed-methods study, to understand how, and by whom, EPaCCS are being used and whether EPaCCS are enabling Healthcare Professionals (HCPs) to coordinate patients' EOL care.

**Methods and analysis** This is a mixed-methods study, carried out within a realist paradigm, to evaluate the impact of an EPaCCS on EOL care as provided by a Clinical Commissioning Group (CCG) in England. This study has two aims: (1) Describe the socio-demographic characteristics of patients who die with an EPaCCS record, their underlying cause of death and place of death and compare these with patients who die without an EPaCCS record. (2) Explore the impact of an EPaCCS on the experience of receiving EOL care for patients and their carers, and understand HCPs' views and experiences of utilising an EPaCCS to coordinate care for their patients. The study will be conducted in five phases: (1) development of the initial programme theory; (2) focus group with CCG stakeholder board; (3) individual interviews with HCPs, patients, current and bereaved carers; (4) retrospective cohort study of routinely collected data on EPaCCS usage and (5) data analysis and synthesis of study findings.

**Ethics and dissemination** The study has been approved by National Health Service South West–Frenchay Research Ethics Committee (REC reference number: 18/SW/0198). Findings will be published in a wide range of outputs targeted at key audiences.

## Strengths and limitations of this study

► Using a theory-driven realist evaluation approach, findings from this study are expected to generate contextually relevant evidence for other care coordination systems.

► The study will test and refine these theories using a mixed methods approach, enhancing the credibility of the evaluation findings.

► This study addresses the need for qualitative research into the use of EPaCCS, offering much needed insight into patient and carers' experiences of EPaCCS.

► The study will investigate the impact of only one of the many EPaCCS developed and implemented in the UK.

► The qualitative component has a potentially small sample size; however, the aim of the study is not to find a robust causal mechanism for EPaCCS, but to unpack the contexts and mechanisms that work in certain circumstances.

## BACKGROUND

People at the end-of-life (EOL; taken here to mean in the last 12 months of life) frequently receive care from a wide variety of teams and organisations. Much of this care is accessed in the out-of-hours period (overnight and at the weekend), when they are unlikely to see a healthcare professional (HCP) who knows them well. Out-of-hours provision of palliative care (defined by the National Institute for Health and Clinical Excellence (NICE) as 'the active holistic care of patients with advanced, progressive illness'[1]) has been identified as a key priority for future research by the Palliative and End of Life Care Priority Setting Partnership, initiated in 2013 by Marie Curie.[2] This process identified the top 10 unanswered research questions and the question with the highest priority was, 'What are the best ways of providing palliative care outside of working hours to avoid crises and help patients to stay in their place of choice?'

*Context*

Something that existed prior to the introduction of the EPaCCS, for example cultural views and beliefs around talking about death and dying and Advanced Care Planning (ACP) conversations. Contexts might also refer to the setting, (e.g. primary or secondary care) or patient characteristics (e.g. underlying diagnosis, socio-demographics, mental capacity).

*Mechanism*

The intended or unintended resources created by an intervention and the response to those resources (cognitive, emotional, motivational etc) by Healthcare Professionals (HCPs), patients and carers. Mechanisms can pertain to why HCPs and patients choose (or choose not) to utilise the EPaCCS.

*Outcome*

An outcome will define the result of the EPaCCS whether intended (did the project succeed against the criteria it set itself at the outset), and also the unplanned and/or unexpected impacts.

Informed by (24) and (26).

**Figure 1** Definition of context, mechanism and outcome.

Continuity of care is important for anyone with complex health and social care needs, but particularly for those at EOL.[3] Until recently HCPs have communicated patients' EOL care plans, to other HCPs, by means of a variety of methods, including shared EOL care registers, letters, faxes and telephone and/or face to face conversations. Despite this, a lack of information sharing has been repeatedly cited as a barrier to the provision of good quality EOL care outside of normal working hours.[4–6] Recent studies looking at the experiences and needs of people seeking palliative care out-of-hours found that most patients expect the HCP to be able to access a summary of their complex medical history and many voiced concerns that their full record could not be accessed by out-of-hours clinicians.[5 7] For some patients, lack of access to their notes is a deterrent to accessing care out-of-hours.[6]

Nationally, the policy drive to address the issue of information continuity has resulted in the development of Electronic Palliative Care Coordination Systems, or EPaCCS.[5 8–10] These records are usually completed by the patients' general practitioner (GP) or community nurse, including patients' advance care wishes, and are accessible across multiple care sectors.[10] The purpose of EPaCCS is to provide a shared local record for health and social care professionals, with key information about an individual approaching the EOL, including their expressed preferences for care. In accordance with the Quality Statements in the NICE End of Life Care Standard for Adults,[11] the intent is for EPaCCS to support people approaching the EOL to receive consistent care that is coordinated effectively across all relevant settings.

An EPaCCS record can take various forms, including web-based electronic registers, systems based on sharing care summaries or care plans, alongside patients' electronic records. They store a dynamic record of a patient's medical condition, treatment, wishes and preferences, and provide information about the medication a patient is receiving, contact details of any carers and services involved in providing care and support to the patient.

Sharing information about patients' EOL care has the potential to improve coordination and communication across care settings.[12] It may reduce the chance of emergency department attendance, hospital admission and dying in hospital.[8 13]

It is now recognised that place of death is unlikely to be the most important factor in achieving a good death and a recent UK study has proposed that it is the presence of loved ones that is more important than the physical location.[14] However, death in preferred place remains a significant measure of quality of death[15] and, according to the Voices survey of bereaved people, despite 81% of respondents believing that the deceased had wanted to die at home less than a quarter of people actually achieve this.[16 17]

Quantitative studies have shown striking differences in place of death with EPaCCS but are potentially biased and confounded.[12] A more recent study challenges these

**Table 1** Programme theory for the EPaCCs study, comprising the 17 CMO statements that inform the programme theory, and the questions that will be used, in the focus group with the end-of-life board, to investigate each CMO statement

| EPaCCs process | CMOs | Focus group questions |
|---|---|---|
| **Commissioning** | 1. If the strategy behind the EPaCCs is definable, deliverable and measurable, the aim, purpose and outcomes of EPaCCS will be clear. *(strategy)* | ▶ How will you evaluate EPaCCS success? <br> ▶ What are the markers of success for you? <br> ▶ What is the CCGs long term vision for the EPaCCS? |
| | 2. If HCPs engage with the EPaCCS positively on early usage and see it as an improvement on any previous EOL register, HCPs will engage positively with EPaCCS. *(engagement)* | ▶ Given that the previous EOL register was generally not well thought of, or used, how did the CCG plan to get HCPs on board? <br> ▶ How do you think the EPaCCS has been received? |
| **Commissioning/uptake/adoption** | 3. If the EPaCCS is well-publicised and marketed to all stakeholders HCPs will be aware of EPaCCS, understand the aim and purpose of the EPaCCS, and will initiate an EPaCCs template and/or access an EPaCCS record. *(publicity)* | ▶ How was the EPaCCS publicised and marketed to different groups of HCPs? What are your views on how effective this has been? <br> ▶ How aware do you think HCPs are of EPaCCS and do you think they understand its purpose and importance? |
| | 4. If HCPs receive sufficient support and training, so that they know how to use it, they and will initiate an EPaCCs template and/or access an EPaCCS record. *(training)* | ▶ Can you tell us about the CCG strategy for providing training and support to different groups of HCPs in the EPaCCS roll-out? <br> ▶ What do you think about this, and how effective it has been? |
| **Uptake/adoption** | 5. If HCPs have the time and/or resources to learn a new system, an EPaCCS template will be initiated. *(time and resources)* | ▶ There are a significant number of GP practices that have not initiated an EPaCCS – do you have any thoughts about why this might be? <br> ▶ Do you think all HCPs will have the time and resources (ie, they are connected to a computer, have internet and NHS network access) to learn and new system and access EPaCCS? |
| | 6. If HCPs are incentivised to use EPaCCS, an EPaCCS template will be initiated. *(incentives)* | ▶ Do HCPs have other ways of obtaining the information contained on EPaCCs? <br> ▶ What might these be, and are these ways better or worse, more reliable or less reliable? |
| **Uptake** | 7. If the patient consents to information-sharing and storage of information about their care preferences, an EPaCCS template will be initiated. *(information sharing)* | ▶ For the EPaCCS to be effective, patients must consent to information-sharing, and the storage of information. <br> ▶ Did you anticipate that this would raise any issues? |
| | 8. If HCPs are near to a computer, are connected to the internet and have access to the GP EMIS Web record, an EPaCCS template will be initiated. *(access to system)* | ▶ There is a theory that because EPaCCS is an electronic record, presently only updateable by the GP on EMIS Web, that this will have an impact on the ability of others to access it and update it and own it. Do you see this as an issue? <br> ▶ What impact do you think this might present? |
| | 9. If HCPs feel able/comfortable having ACP conversations with patients, an EPaCCS template will be initiated. *(ACP conversations)* | ▶ How do you think HCPs feel about having ACP conversations with patients? <br> ▶ Research suggests that patients with non-malignant diagnoses are less likely to be added to EPaCCS. <br> ▶ Do you think this is the case and if so why? Are there other patient groups who might be under-represented on the EPaCCS? |
| | 10. If HCPs feel that the EPaCCS facilitates, potentially difficult, ACP conversations an EPaCCS template will be initiated. *(ACP conversations)* | ▶ Some would argue that the EPaCCS template might facilitate ACP conversations with patients–what are your thoughts on this? |
| | 11. If the patient is willing, and has capacity to have ACP conversations, an EPaCCS template will be initiated. *(ACP conversations)* | ▶ Patients can only record their wishes if they are able to have a conversation with an HCP–what issues do you think this might present? |
| **Adoption** | 12. If End of Life Care information about a patient can be accessed more efficiently in other ways (ie, speaking with carer or reading other sources of information) the information on the EPaCCS template may not be accessed. *(single point of access)* | ▶ Are there any other sources of information that HCPs might access to establish the EOL wishes and needs of a patient and do you think they present an issue of the uptake of EPaCCS? |
| | 13. If HCPs are near to a computer, are connected to the internet and have access to the NHS Network an EPaCCs template will be accessed. *(access to system)* | ▶ There is a theory that because EPaCCS is an electronic record, presently only updateable by the GP on EMIS Web, that this will have an impact on the ability of others to access it and update it and own it. <br> ▶ Do you see this as an issue? What impact do you think this might present? |

Continued

**Table 1** Continued

| EPaCCs process | CMOs | Focus group questions |
|---|---|---|
| **Adoption/care coordination** | 14. If the information does not reflect the current wishes of the patient, care may not be aligned with the patients' preferences. *(patient preferences)* | ▶ Do you feel that the EPaCCs adequately reflects patient's wishes and preferences for care? |
| | 15. If the patient does not have clear or clinically meetable preferences, or their wishes are subject to frequent change, care may not be aligned with the patient's wishes. *(patient preferences)* | ▶ Do you feel the EPaCCS adequately reflects the patient's/carer's wishes and preferences regarding end of life care and do you feel these wishes are deliverable? If not, why might this be and what needs to be improved? |
| **Care coordination** | 16. If HCPs access EPaCCS and consider the information contained within it to be trustworthy (current, relevant, detailed and useful) care will be coordinated by EPaCCs and this care will align with the patient's wishes. *(trustworthiness of EPaCCS)* | ▶ Do you think the EPaCCS contains all the information HCPs need to enact their patient's wishes and coordinate their patient's care?<br>▶ Do you consider it to be current, relevant, detailed and useful? If not, why might this be and what needs to be improved? |
| | 17. If EPaCCS does not enhance or improve the care that is already being delivered care may not be coordinated by EPaCCS, consistent or reflect the patients' preferences. *(patient outcomes)* | ▶ What are your thoughts on the notion that: 'The EPaCCS is not coordinating care, it is simply recording what is already being done' |

ACP, advanced care planning; CCG, Clinical Commissioning Group; CMO, context, mechanism and outcome; EOL, end of life; EPaCCS, electronic palliative care coordination systems; GP, general practitioner; HCP, healthcare professional; NHS, National Health Service.

assumptions, suggesting that the increase in home deaths could in fact be due to selection bias (few secondary care colleagues used the systems and therefore hospital deaths are not captured).[10] The findings of this study also underscore the importance of qualitative approaches, which can offer crucial insights into what is happening on the ground, away from broad claims of EPaCCS benefits arrived at solely through quantitative methods. Without understanding the experiences of patients and carers, together with the perspectives of HCPs, it is difficult to evaluate the effectiveness of EPaCCS.[18]

Technology in isolation is not guaranteed to bring benefit and the initiation of an EPaCCS relies on HCPs initiating conversations about death and dying. There is evidence that these conversations are difficult for HCPs,[19] with many choosing to avoid the conversation altogether.[20] What impact EPaCCS has on these conversations, if any, is not clear.

A recent independent evaluation of EPaCCS found that 'it was not possible to demonstrate that EPaCCS was making any difference to the care patients were receiving at EOL because the range of clients for whom EPaCCS was being used remains focused on cancer, and the ability of EPaCCS systems to report on progress and outcomes remains generally poor.'[21]

We therefore do not know if EPaCCS acts to improve practice or whether it simply documents and reflects what is already taking place in practice.[21] Indeed, the need to gather evidence of effectiveness of EPaCCS is vitally important, as it has already been widely and uncritically adopted by the National Health Service (NHS).

To summarise, very little research has been carried out to understand how, and by whom, EPaCCS are being used and, perhaps more importantly, whether EPaCCS are enabling HCPs to support patients' EOL wishes. Rigorous evaluation and research are urgently needed to investigate

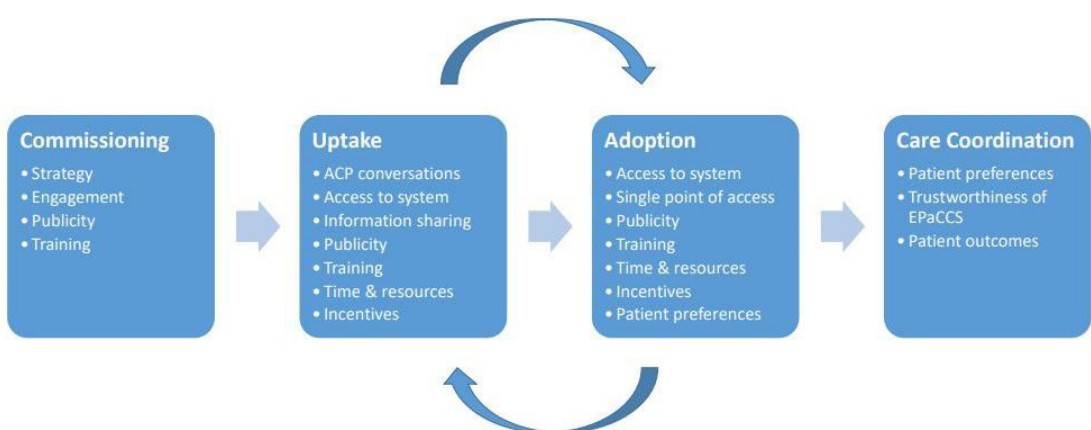

**Figure 2** Visual representation of the initial programme theory at a macro level. ACP, advanced care planning; EPaCCS. electronic palliative care coordination systems.

to what extent EPaCCS influence services working together to support 'a good death', the outcome that stakeholders think is of most importance.[22]

## AIMS, OBJECTIVES AND RESEARCH QUESTIONS

The study has two aims. These are to:

1. Describe the socio-demographic characteristics of patients who die with an EPaCCS record, their underlying cause of death and place of death and compare these with patients who die without an EPaCCS record.
2. Explore the impact of an EPaCCS on the experience of receiving EOL care for patients and their carers, and understand HCPs' views and experiences of utilising an EPaCCS to provide palliative care to their patients.

## PROJECT METHODOLOGY
### Study setting

This study will be implemented in England, where one Clinical Commissioning Group (CCG) has recently developed and rolled out an EPaCCS. This involved the dissemination of an EMIS Web template, which was circulated to general practices, to help ensure consistent data entry and coding. EMIS Web is the most widely used primary care clinical system in the UK and allows real-time patient information to be shared securely between different organisations. All practices in the CCG area are EMIS Web users and it is now also used by the local hospice and many of the community nursing teams.

The EPaCCS template was developed following extensive local clinical consultation and the National Information Standard.[23] Although some organisations within the CCG area (ambulance service, secondary and social care) are non-EMIS Web users, information from the EPaCCS template can be viewed across the local health community, via the integrated digital care record used by health and social care professionals in the CCG area which went live at the end of February 2018. The integrated digital care record contains some of the information held at GP practices, hospital departments, community services, mental health trusts, out of hours services and local authorities across the CCG area, combining it into a single, shared digital record.

### Conceptual framework

This study will draw on a realist evaluation approach.[24] A randomised, experimental study design is not possible as the implementation of EPaCCS has been strongly advocated and promoted by NHS England, with 83% of CCGs in England reported to have an operational EPaCCS, or be in the planning stages, by 2013.[25] The CCG had recently operationalised an EPaCCS at the time of study design.

By their nature, EPaCCS involve multiple organisations and a multidisciplinary style of work. It therefore requires a novel methodological approach to evaluation as described in this protocol.

Realist evaluation is a theory-driven approach designed for evaluating complex interventions, such as EPaCCS, where the outcomes are influenced by the way the intervention is delivered and in what context. Due to this complexity, any evaluation of EPaCCS seeking to determine linear causal relationships or simply find out if the intervention 'works' is unlikely to be useful. Pawson and Tilley,[24] the developers of realistic evaluation, suggest that the results of an intervention (outcomes) are dependent on the introduction of appropriate reasoning and resources (mechanisms) and how these then interact with existing social and cultural condition (contexts). For the purposes of this study, we have defined context (C), mechanism (M) and outcome (O) in figure 1.

A realist approach acknowledges that complex interventions only ever work for certain people, in particular circumstances. The key task of a realist evaluation is to understand and explain these patterns by asking the exploratory question: *what works, for whom and in what circumstances?*

According to Pawson, interventions or 'programmes', such as the EPaCCS, are 'theories incarnate' and every programme has a theoretical underpinning.[26] Before conducting a realist evaluation, the researchers must develop a theory, or theories, that explain what works, for whom, under what circumstances and how. This is sometimes known as the 'initial programme theory' and is described through CMO conjectures. This theory, or theories, are then tested through the process of the evaluation.

The study will be conducted in five phases: (1) development of the initial programme theory; (2) focus group with CCG stakeholder board; (3) individual interviews with HCPs, patients, current and bereaved carers; (4) retrospective cohort study of routinely collected data on EPaCCS usage and (5) data analysis and synthesis of study findings.

### Phase 1: development of the initial programme theory (June–October 2018)

Phase 1 is complete and included identification of relevant literature from electronic searches of databases, such as Medline and Google Scholar. The search strategy involved searching for papers which discussed or evaluated shared digital records, for the coordination of palliative care, EOL care or advance care plans. Reference lists of relevant papers were scanned, and citation searches conducted. Grey literature relating to policy and organisational-based material were sought by searching government and other specialist websites. The lead researcher's own experiences as a GP were used as 'informed guesswork'[27] and initial meetings were held with key stakeholders, including patients at the local hospice and commissioners. These initial engagements were informal and were patient and public involvement activities (detailed further below). They did not constitute formal interviews requiring ethics clearance.

The proposed implementation of EPaCCs was broken down and analysed, to understand different elements of this process (table 1, column 1). These elements, highlighted as important through literature searches and initial

stakeholder engagement, were analysed and detailed into initial CMO statements (table 1, column 2). An overview of these CMO statements were then illustrated through a process diagram as illustrated in figure 2.

At an early stage of the programme theory development, the CMO conjectures were reviewed by, and discussed with, Dr Justin Jagosh, Director of the Centre for Advancement in Realist Evaluation and Synthesis.

The initial programme theory forms a set of hypotheses on what the mechanisms may be, what groups may benefit most or least and the contextual factors that might be important to its success or failure. These hypotheses will be interrogated and tested in phases 2–5 of the study.

## Phase 2: focus group with CCG stakeholder board (October 2018)

The CCG EOL care board is a multi-disciplinary, multi-organisational system board, whose members are high-level stakeholder representatives from across the CCG, including representatives from community nursing teams, primary care, the ambulance service (which serves a wider geographical area than the CCG area), local hospices, care homes and secondary care. The purpose of the board is to review the current commissioning across all geographical provision, ensuring unified pathways for community, primary and secondary providers to provide consistency for all patients, carers and staff across the system.

Board members will be invited to take part in a focus group following their attendance at the monthly board meeting. Focus groups allow for social interaction which can help to reveal issues and subsequent points of view that may not be prompted or discovered through individual interviews. This approach will help the study team to gain as wide an understanding as possible of the process of commissioning the EPaCCS and help to refine the initial programme theory.

Participants will be consented to take part in the study prior to the focus group by either LF or LP who will be facilitating. A topic guide will be used to steer the focus group and will enable the research team to test and refine the initial programme theory prior to commencing the in-depth interviews (see table 1).

It is anticipated that the focus group will last approximately 45 min and that approximately 5–10 participants will take part in it. The focus group will be audio taped and transcribed verbatim.

## Phase 3: individual interviews with HCPs, patients, current and bereaved carers (November 2018–July 2019)
### Healthcare professionals

HCPs from community nursing teams, primary care, the ambulance service, the local hospices, care homes and secondary care will be invited to participate.

GPs working in practices within the CCG area will be invited to support the study and the study team will purposively sample from a list of practices, to include practices that are high-users of EPaCCS and low users of EPaCCS based on data compiled by the CCG.

The research team will purposively sample HCPs who express an interest in participating according to gender, age and profession to ensure maximum variation in the sample. All interviews with HCPs will take place over the telephone for both pragmatic and methodological reasons. Conducting interviews over the telephone will reduce the time and cost to the study that may be involved in travelling to interviews and well-planned telephone interviews can gather the same material as those held face to face.[28]

A topic guide, informed by the evolving programme theory, will be used to ensure consistency across the interviews. This will enable the research team to compare the views of each group at the stage of data analysis. Interviews will last approximately 30 min and it is anticipated that approximately 3–5 HCPs will be interviewed from each group (18–30 in total).

### Patients, current and bereaved carers

Patients will be approached to take part in interviews through their GP surgeries or the local hospice. The research team will purposively sample from a list of practices, to include practices that are high-users of EPaCCS and low users of EPaCCS. High EPaCCS use will be defined as practices that have created greater than 20 EPaCCS records (the median number of records across all practices) 4 months post-implementation, based on data extracted by the CCG. Selected practices will be asked to identify patients, aged 18 and over, receiving EOL care, who they consider might be eligible to take part in the study.

### Inclusion criteria

1. Capacity to give informed consent.
2. Aged 18 and over.
3. Prognosis 12 months or less as identified by their GP (patient aware of this prognosis).

Potential participants will be sampled purposively to include patients across the age range, with and without an EPaCCS record, with both malignant and non-malignant health conditions.

To recruit patients to the study from the local hospice we will liaise with key clinical staff, who will be responsible for identifying appropriate patients. Once again, potential participants will be sampled purposively to include patients across the age range, with and without an EPaCCS record, with both malignant and non-malignant health conditions.

Alongside their own study information pack, all patients will be given a carer information pack which they can choose to give to their carer if they are happy for their carer to participate in the study.

GPs will also be asked to identify recently bereaved carers (between 8 weeks and 6 months of the death of their relative), who they consider might be eligible to take part in the study. GPs will be sent details of how to perform an appropriate search within EMIS Web to identify potential participants.

Interviews with carers will be conducted one-to-one with the interviewer. Interviews with patients will be conducted

either one-to-one with the interviewer or in the presence of their carer, according to the wishes of the patient.

Interviews will last approximately 45 min and it is anticipated that approximately 15 patients will be interviewed (10 EPaCCS patients and 5 non-EPaCCS patients) and 10 carers (to include both current and bereaved). All interviews will be audio taped and transcribed verbatim.

## Phase 4: retrospective cohort study of routinely collected data on EPaCCS usage (March 2019–July 2019)

EMIS Web data will be accessed to identify all patients, aged 18 and over, who die in the CCG area between 22 February 2018 and 21 February 2019. Agreements are in place with the CCG to obtain this data. Patients will be identified as either having an EPaCCS record (EPaCCS patient) or not (non-EPaCCS patient), using EMIS Web coding.

EMIS Web will be used to characterise both EPaCCS and non-EPaCCS patients in terms of their gender, ethnicity and postcode (as a proxy for socio-economic status according to their Index of Multiple Deprivation Score). Data will be extracted to describe:

1. The proportion of patients that die with an EPaCCS record.
2. When the EPaCCS record is initiated (ie, how many months/days prior to the patient's death), and by whom.
3. How frequently the EPaCCS record is updated once opened, and who makes any changes.
4. The underlying cause and place of death for EPaCCS and non-EPaCCS patients.
5. The number of hospital admissions and Emergency Department attendances for EPaCCS and non-EPaCCS patients in the last 12 months of life.

Descriptive data will be collected, by the CCG, on EPaCCS usage, including the number of records created by each GP surgery in the CCG area. Data from the integrated digital care record will be accessed to describe which HCPs (GPs, community nurses, hospice HCPs, ambulance HCPs and secondary care clinicians) are accessing these shared EOL care records.

## Phase 5: data analysis and synthesis of study findings (October 2018–October 2019)
### Quantitative methods

Quantitative data will be analysed using Stata V.15 and reported using descriptive statistics. Within the context of this realist evaluation, we were keen to use the quantitative data to address a single hypothesis, namely whether nominal possession of an EPaCCS record was associated with increased chance of dying at home. However, logistic regression will be used to determine the adjusted OR and 95% CI for the associations between having an EPaCCS record and dying at home, considering other factors of interest, including, but not limited to: age, sex, deprivation and underlying cause of death.

Of approximately 8000 deaths occurring in the CCG area over the year of study, we expect around 10% (800

deaths) of patients to have an EPaCCS.[10] If the proportion of deaths occurring at home is expected to be 25% among those without an EPaCCS, we would have over 99% power to detect an absolute increase of 10% to 35% among those with an EPaCCS. The power would be about 84% if the proportion were increased by 5% to 30%.

Descriptive statistics will be employed to report EPaCCS usage.

### Qualitative methods

Data analysis will be conducted using a realist approach informed by Jackson and Kolla's realistic evaluation analysis method.[29] This analytic process will involve the following steps:

1. Coding individual units (a discrete C, M or O) within the narratives of the interviews.
2. Identifying the complex connections that link these codes together into dyads or triads.
3. Subsuming the linked codes into themes using thematic analysis.[30]

Analysis will begin shortly after data collection starts and be ongoing and iterative. Analysis will inform further data collection: for instance, analytic insights from data gathered in earlier interviews will be used to develop and adapt the programme theory and in turn, identify any changes that need to be made to the topic guide for use during later interviews. The study will generate new programme theories to explain how the EPaCCS works, for whom and any contextual influences and constraining factors that affect their initiation and usage. Emerging analysis and findings will be discussed with PPI representatives, to explore and clarify findings.

Qualitative and quantitative data will be collected concurrently, giving equal weight to each.[30] Data will be triangulated in order to test and refine the programme theories, accepting that any findings are fallible and with time and further study new data are bound to emerge.[31] The synthesised study findings will establish the potential outcomes of EPaCCS, identify the underlying mechanisms which explain how they produce these effects and highlight the key contextual factors that affect their success or failure. Recommendations can then be made for the development and implementation of EPaCCS.

### Patient and public involvement

To support the development of this study protocol, members of the study team (LF and LP) presented and discussed an outline proposal of this study to patients, staff and carers at the local hospice on two separate occasions in April 2018. Approximately 10 participants voluntarily took part in these, semi-structured, discussions, in which we asked specific questions concerning ethical and methodological issues. Participants were also encouraged to ask any questions. These meetings raised several important issues which have been incorporated into the design of this study. Such issues included allowing patients the choice of whether to have a carer sit alongside them during their interview and which HCPs they

felt it was important that the study team spoke to, due to the involvement they had in providing care for patients. The meetings also discussed what terms, wording and questions would be acceptable to patients and carers to read and hear in the study information documents and interviews.

At the end of both meetings, patients and carers were invited to continue to support the design of the study should they wished. Two members came forward expressing a wish to be more actively involved in the study. They have kindly been involved in reviewing all the study literature, including the topic guides, study information sheets and the lay summary for this protocol. It is hoped that they will wish to continue their involvement with the study. This will include informing the content of materials for lay audiences, drawing links to groups and forums that the research team may be unaware of, and supporting the study team with the interpretation and dissemination of study findings. To ensure ongoing PPI, all patients and carers taking part in the study will be invited to support the ongoing development of the study.

## Ethics and dissemination

The study has been approved by NHS South West–Frenchay Research Ethics Committee (REC reference number: 18/SW/0198). The research team will disseminate the findings to a range of stakeholders. We will draw on the networks and expertise of the local CCG EOL care board to disseminate the research outputs widely and appropriately. Key audiences include patient and carer organisations, GPs and community nursing teams in primary care, ambulatory services and care home staff, HCPs working in secondary palliative care services and hospices, managers and directors within healthcare organisations with responsibility to provide high-quality services within budget and healthcare policy-makers, nationally and internationally.

**Acknowledgements** We would like to thank all the patients, carers and HCPs who helped to support the PPI in development of this study. We would also like to thank Dr Justin Jagosh, Director of the Centre for Advancement in Realist Evaluation and Synthesis (https://realistmethodology-cares.org/), for his support and guidance in developing the initial programme theory.

**Contributors** LP was the principal investigator for this qualitative study. LF, MF, RM and SP all contributed to the development of the study protocol. LP and LF drafted this manuscript. All authors contributed to the editing of the final manuscript and the refining of its intellectual content.

**Funding** LP's time is paid for by a NIHR School for Primary Care Research GP Career Progression Fellowship. This fellowship funding will also meet the research costs. LF's time is paid for by NIHR Research Capability Funding. MF's time is supported by the NIHR Collaboration for Leadership in Applied Health Research and Care West (NIHR CLAHRC West). The study is supported by the NIHR West of England Clinical Research Network. The views expressed in this publication are those of the authors and not necessarily those of the NHS, the National Institute for Health Research or the Department of Health and Social Care.

**Competing interests** None declared.

**Patient and public involvement** Patients and/or the public were involved in the design, or conduct, or reporting, or dissemination plans of this research. Refer to the Methods section for further details.

**Patient consent for publication** Not required.

**Provenance and peer review** Not commissioned; externally peer reviewed.

**Data availability statement** Data sharing not applicable as no datasets generated and/or analysed for this study.

**ORCID iD**
Lydia French http://orcid.org/0000-0002-8517-2148

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
