## [Reviewer comments · BMJ Open]

ARTICLE DETAILS

TITLE (PROVISIONAL)	THE IMPACT OF ELECTRONIC PALLIATIVE CARE COORDINATION SYSTEMS (EPaCCS) ON CARE AT THE END OF LIFE ACROSS MULTIPLE CARE SECTORS, IN ONE CLINICAL COMMISSIONING GROUP AREA, IN ENGLAND: A REALIST EVALUATION PROTOCOL
AUTHORS	Pocock, Lucy; French, Lydia; Farr, Michelle; Morris, Richard; Purdy, Sarah

VERSION 1 - REVIEW

REVIEWER	Pablo Millares Martin Whitehall Surgery, Leeds, UK
REVIEW RETURNED	17-May-2019

GENERAL COMMENTS	There is a limitation in the study which does not seem to be given the right relevance. The study is going to be carried out among EMIS organisations, and although it covers primary care well it does not involve the whole spectrum. It is stated ambulance service -quite relevant in this matter-, social care and secondary care do not use it. In consequence, interoperability gap is not addressed. Regarding phases 2 and 3 of the study, I wonder whether there is a going to be mixed approach, allowing during the process information from health care professionals/patients/bereaved feedback/influence the questions to other groups and CCG stakeholders and vice versa. It would be more powerful than having separate bodies of opinion that could clearly diverge. In phase 5, there is too much focus on EPaCCS patients. If the aim is also to consider non-cancer patients, there is a need to assess why patients with long term conditions were not on EPaCCS, and then to consider how many deaths could be considered expected, without EPaCCS, and how many were unexpected/requiring coroner involvement. Statistics section is quite short, and unfortunate to give the impression only considering "associations between having an EPaCCS record and dying at home". There is much more to the need to be explored, and more exploration on factors like level of usage/level of coverage among organisations and also as regards as factors that impeded the good death, and not seen only as dying at home, but dying with family and relatives present, and symptom free.
---

REVIEWER	Claire Creutzfeldt University of Washington
REVIEW RETURNED	15-Jun-2019

GENERAL COMMENTS	I think we need a little more description of what is currently standard and what is not. The EPaCCS is already established and will not be changed, although needs to be further disseminated.
--

REVIEWER	Matthew Allsop University of Leeds, UK
REVIEW RETURNED	18-Jun-2019

GENERAL COMMENTS	Thank you for the opportunity to review your manuscript. Please find my comments regarding specific sections of the manuscript below. Strengths and limitation of study – your first point, about articulating preliminary theories and assumptions about EPaCCS, may require revision to be specific to the region in which the EPaCCS you are examining is based. I do not think you will be clearly articulating preliminary theories and assumptions about all EPaCCS in your project. Background: Paragraph starting ‘Sharing information about patients’ EOL care...’ – I am not sure that Petrova et al supports this statement and would advise reframing the statement or identifying a different source. Similarly, it would be helpful to see specific evidence cited to support the second sentence in the paragraph regarding specific benefits intended from sharing information rather than a strategy document. If the EPaCCS implementation in the CCG of interest has not been rolled out to include secondary care organisations, the ambulance service or providers of social care, please clarify if you expect to obtain a valid representation of the intended implementation of EPaCCS with your proposed project. The focus on one CCG will allow detailed exploration, but how do you plan to translate your findings elsewhere to other regions? Please provide some insight into how representative the EPaCCS you are evaluating is compared to other systems implemented across the UK. Methods: More detail is required for phase one, particularly given the importance of this phase to subsequent research activities. In particular the methods used for identification of relevant literature need to be detailed. It would be helpful to include a copy of a search strategy and PRISMA. I would also encourage reflection on how the findings of the literature identified in this phase relate to a recent review of the literature on evidence for EPaCCS:
--

<https://spcare.bmj.com/content/early/2019/05/08/bmjspcare-2018-001689>

Were your findings similar?

I am also unclear on the process for developing the initial programme theory. More detail is required to determine how the findings presented in Figure 2 were derived from the literature and stakeholder engagement.

How were public and patient involvement activities organised? Were these structured, what was the topic guide or structure of sessions?

The geographical remit of the CCG board is unclear. Which areas are the board responsible for and can you be more specific about the providers who have representation on the board? For example, does the ambulance service only cover the region of the CCG or extend beyond wider regions? This is important in considering the context in which the system is being implemented.

For HCP interviews, how will you determine high and low users of EPaCCS based on data compiled by the CCG? This is essential for understanding the methods surrounding the HCP and patient interviews.

What do you anticipate will be the impact of telephone interviews with HCPs and can you reflect on that in the protocol?

How was sampling of HCPs informed? You will involve 3 – 5 HCPs from across the whole CCG. How many practices are there in the CCG? Can you be confident that you will capture a wide enough range of perspectives from both high and low users reflecting the spread of use of EPaCCS across the whole CCG with 3 – 5 interviews? What is your sampling frame?

Patient interviews – can you determine prognosis of the patient without using an EPaCCS record? Given EPaCCS is the focus on the research, I wondered if this was being assessed independently, and how.

What do you aim to obtain from patient and caregiver interviews? Do you have a topic guide you could include alongside the protocol?

Phase 4 – can you clarify in the protocol whether data has already been secured or whether this is to be arranged? Can further detail be provided in the characterisation of EPaCCS and non-EPaCCS patients? How will you determine this?

Please provide further detail on the integrated digital care record – what is this and how will it enable you to determine who is reviewing EPaCCS records? Why are ambulance HCPs and secondary care included in this list if EPaCCS has not been rolled out to these groups yet? Would you expect them to be accessing EPaCCS?

Phase 5 – on what basis have you determined the expected number of deaths (10%) on EPaCCS? This needs clarification if forming the basis of your power calculation.

	Qualitative methods – will all findings be discussed with PPI representatives to explore and clarify findings – what contributions do you anticipate from the representatives regarding, for example, HCP use (or lack of) by secondary care staff? Dissemination – do you anticipate the value of the research to extend beyond service improvement within the CCG? If so, what pathways to impact do you plan to pursue and how will findings from the study be made relevant for a wide range of audiences?
--	--

VERSION 1 – AUTHOR RESPONSE

Reviewer 1

There is a limitation in the study which does not seem to be given the right relevance. The study is going the carried out among EMIS organisations, and although it covers primary care well it does not involve the whole spectrum. It is stated ambulance service -quite relevant in this matter-, social care and secondary care do not use it. In consequence, interoperability gap is not addressed.

We thank the reviewer for his comments and we agree, it is indeed a limitation of this EPaCCS intervention that social care and secondary care cannot access EMIS and this is something that we expect to find in our data. The interoperability gap is addressed in our initial programme theory (see table 1) and we plan to explore it in detail in the qualitative interviews, which will include members of the ambulance service and secondary care.

Regarding phases 2 and 3 of the study, I wonder whether there is a going to be mixed approach, allowing during the process information from health care professionals/patients/bereaved feedback/influence the questions to other groups and CCG stakeholders and vice versa. It would be more powerful than having separate bodies of opinion that could clearly diverge.

We plan to take an iterative approach, with data collected from one group helping to refine the programme theory and generate questions for the topic guides for later interviews with different groups of participants. This has been deliberately planned so that the data are collected initially from the commissioners, and then the healthcare professionals, and lastly the patients and carers. The refined programme theory presented at the end of the study will represent a synthesis of both the qualitative and quantitative data.

In phase 5, there is too much focus on EPaCCS patients. If the aim is also to consider non-cancer patients, there is a need to assess why patients with long term conditions were not on EPaCCS, and then to consider how many deaths could be considered expected, without EPaCCS, and how many were unexpected/requiring coroner involvement.

The focus on EPaCCS patients is appropriate as this is a realist evaluation of an EPaCCS and we need to understand if, and how, patients with an EPaCCS record differ from those without. We also need to explore the differences in the care that they receive. This is why we will be interviewing patients with, and without, an EPaCCS record. The interviews will also explore the process by which healthcare professionals identify patients for the EPaCCS, and why this might not include patients with non-malignant disease.

Statistics section is quite short, and unfortunate to give the impression only considering "associations between having an EPaCCS record and dying at home". There is much more to the need to be explored, and more exploration on factors like level of usage/level of coverage among organisations and also as regards as factors that impeded the good death, and not seen only as dying at home, but dying with family and relatives present, and symptom free.

We agree that there is more work to be done in this area. We appreciate that dying at home should not be the only marker of a good death and we will be exploring many of the other factors mentioned within the qualitative interviews. We have amended the statistics section on page 15 so that it takes account of this issue. The text now reads:

“Quantitative data will be analysed using Stata v15 and reported using descriptive statistics. Within the context of this realist evaluation, we were keen to use the quantitative data to address a single hypothesis, namely whether nominal possession of an EPaCCS record was associated with increased chance of dying at home. However, logistic regression will be used to determine the adjusted odds ratio and 95% confidence intervals for the associations between having an EPaCCS record and dying at home, considering other factors of interest, including, but not limited to: age, sex, deprivation and underlying cause of death.

Of approximately 8,000 deaths occurring in the CCG area over the year of study, we expect around 10% (800 deaths) of patients to have an EPaCCS (10). If the proportion of deaths occurring at home is expected to be 25% among those without an EPaCCS, we would have over 99% power to detect an absolute increase of 10% to 35% among those with an EPaCCS. The power would be about 84% if the proportion were increased by 5% to 30%.

Descriptive statistics will be employed to report EPaCCS usage.”

Reviewer: 2

I think we need a little more description of what is currently standard and what is not. The EPaCCS is already established and will not be changed, although needs to be further disseminated.

We thank the reviewer for her comments. To provide a little more description of what is currently standard and what is not we have added the following text to the Introduction on page 4:

“Until recently HCPs have communicated patients’ end of life care plans, to other HCPs, by means of a variety of methods, including shared end of life care registers, letters, faxes and telephone and/or face to face conversations. Despite this, a lack of information sharing has been repeatedly cited as a barrier to the provision of good quality EOL care outside of normal working hours (4,5,6).”

Under the section ‘project methodology’, on page 8, we have also added the following text:

“...with 83% of Clinical Commissioning Groups (CCGs) in England reported to have an operational EPaCCS, or be in the planning stages, by 2013 (25).”

Reviewer: 3

Strengths and limitations of study - your first point, about articulating preliminary theories and assumptions about EPaCCS, may require revision to be specific to the region in which the EPaCCS you are examining is based. I do not think you will be clearly articulating preliminary theories and assumptions about all EPaCCS in your project.

We thank the reviewer for his comments. To make clear that we will not be articulating preliminary theories and assumptions about all EPaCCS we have amended the fourth bullet point under the section ‘strengths and limitations’ so that it now reads:

“The study will investigate the impact of only one of the many EPaCCS developed and implemented in the UK.”

Background:

Paragraph starting 'Sharing information about patients' EOL care...' – I am not sure that Petrova et al supports this statement and would advise reframing the statement or identifying a different source.

We have amended the statement attributed to Petrova et al, so that it now reads:

"Sharing information about patients' EOL care has the potential to improve coordination and communication across care settings (12)."

Similarly, it would be helpful to see specific evidence cited to support the second sentence in the paragraph regarding specific benefits intended from sharing information rather than a strategy document.

To support the second sentence in the paragraph regarding the benefits intended from sharing information, we have added the following references which are not taken from strategy documents alone.

"Sharing information about patients' EOL care has the potential to improve coordination and communication across care settings (12). It may reduce the chance of emergency department attendance, hospital admission and dying in hospital (8, 13)."

If the EPaCCS implementation in the CCG of interest has not been rolled out to include secondary care organisations, the ambulance service or providers of social care, please clarify if you expect to obtain a valid representation of the implementation of EPaCCS with your proposed project.

Under the sub-heading project methodology on page 7, we have added the following text to make clear what access various organisations will have to EPaCCS across the CCG.

"Although some organisations within the CCG area (ambulance service, secondary and social care) are non-EMIS users, information from the EPaCCS template can be viewed across the local health community, via the integrated digital care record used by health and social care professionals in the CCG area, which went live at the end of February 2018"

The focus on one CCG will allow detailed exploration, but how do you plan to translate your findings elsewhere to other regions? Please provide some insight into how representative the EPaCCS you are evaluating is compared to other systems implemented across the UK.

The EPaCCS under study is sufficiently similar to others implemented across the UK, meaning that the contexts, mechanisms and outcomes generated are likely to be applicable to other care coordination systems nationally. We intend to publish the findings of our study in a peer-reviewed journal, where we will address these important issues.

Methods: More detail is required for phase one, particularly given the importance of this phase to subsequent research activities. In particular the methods used for identification of relevant literature need to be detailed. It would be helpful to include a copy of a search strategy and PRISMA. I would also encourage reflection on how the findings of the literature identified in this phase relate to a recent review of the literature on evidence for EPaCCS:

<https://spcare.bmj.com/content/early/2019/05/08/bmjspcare-2018-001689> Were your findings similar?

Thank you for your points, the review of literature in phase one was a pragmatic review, using snowballing and hand-searching methods, rather than a systematic review and is therefore not covered by the PRISMA checklist. This is in line with the RAMESES II guidance on carrying out realist evaluations

(https://www.ramesesproject.org/media/RAMESES_II_Developing_realist_programme_theories.pdf).

At the time of submission, the literature review you cite had not yet been published. We will of course address this in the paper in which we report our findings.

I am also unclear on the process for developing the initial programme theory. More detail is required to determine how the findings presented in Figure 2 were derived from the literature and stakeholder engagement.

To make clear how the findings presented in the Figure 2 were derived from the literature and stakeholder engagement under the subheading; “Phase one: development of the initial programme theory (June-October 2018)”, on page 10, we have added the following text:

“The proposed implementation of EPaCCs was broken down and analysed, to understand different elements of this process (Table 1, Column 1). These elements, highlighted as important through literature searches and initial stakeholder engagement, were analysed and detailed into initial CMO statements (Table 1, Column 2). An overview of these CMO statements were then illustrated through a process diagram as illustrated in Figure 2.”

How were public and patient involvement activities organised? Were these structured, what was the topic guide or structure of sessions?

the section ‘patient and public involvement’, on page 16, we have added the following text to make clear how public and patient involvement activities were organised:

“Approximately 10 participants voluntarily took part in these, semi-structured, discussions, in which we asked specific questions concerning ethical and methodological issues. Participants were also encouraged to ask any questions. These meetings raised several important issues which have been incorporated into the design of this study.”

The geographical remit of the CCG board is unclear. Which areas are the board responsible for and can you be more specific about the providers who have representation on the board? For example, does the ambulance service only cover the region of the CCG or extend beyond wider regions? This is important in considering the context in which the system is being implemented.

In the interests of ensuring the confidentiality of any potential study participants, particularly members of the CCG end-of-life board, we have been unable to say much more about the geographical remit of the CCG or members of the board. We have however added the following text, on page 10, to make clear that the ambulance service does cover an area wider than the CCG.

“The CCG EOL care board is a multi-disciplinary, multi-organisational system board, whose members are high-level stakeholder representatives from across the CCG, including representatives from community nursing teams, primary care, the ambulance service (which serves a wider geographical area than the CCG area), local hospices, care homes and secondary care.”

For HCP interviews, how will you determine high and low users of EPaCCS based on data compiled by the CCG? This is essential for understanding the methods surrounding the HCP and patient interviews.

To make clear that high and low EPaCCS users were identified from the CCG data we have added the following text to the section ‘patients, current and bereaved carers’ on page 12:

“The research team will purposively sample from a list of practices, to include practices that are high-users of EPaCCS and low users of EPaCCS based on data compiled by the CCG which makes clear the number of EPaCCS templates initiated according to practice

What do you anticipate will be the impact of telephone interviews with HCPs and can you reflect on that in the protocol?

We would like to draw the reviewer's attention to the following sentence, page 12, which we feel makes clear any impact of conducting a telephone interview: "All interviews with HCPs will take place over the telephone for both pragmatic and methodological reasons. Conducting interviews over the telephone will reduce the time and cost to the study that may be involved in travelling to interviews and well-planned telephone interviews can gather the same material as those held face to face (29)

How was sampling of HCPs informed? You will involve 3 – 5 HCPs from across the whole CCG. How many practices are there in the CCG? Can you be confident that you will capture a wide enough range of perspectives from both high and low users reflecting the spread of use of EPaCCS across the whole CCG with 3 – 5 interviews? What is your sampling frame?

We would like to draw the reviewer's attention to the following sentence, on page 11, which explains that we plan to interview up to 30 HCPs from different professional groups.

"3-5 HCPs will be interviewed from each group (18-30 in total)."

The study has been designed according to the realist methodology which aims to find an appropriate sample of respondents, who can provide information about contexts, mechanisms and/or outcomes. In her paper on realist interviewing (<https://journals.sagepub.com/doi/10.1177/1356389016638615>), Ana Manzano explains this very nicely: "realist hypotheses are not confirmed or abandoned through saturation obtained in a double-figure number of qualitative interviews but through relevance and rigour (Pawson, 2013) obtained in a mixed-method strategy. A theory may be gleaned, refined or consolidated not necessarily in the next interview, but also while digging for nuggets of evidence (Pawson, 2006) in other sources of data (i.e. documents, routinely collected administrative data)."

Patient interviews – can you determine prognosis of the patient without using an EPaCCS record? Given EPaCCS is the focus on the research, I wondered if this was being assessed independently, and how.

The patients were identified by their GP practice, who will have assessed them to be in the last 12 months of their life, according to the sampling criteria as detailed in the study protocol. We have clarified this by adding the following text "as identified by their GP", to page 12.

What do you aim to obtain from patient and caregiver interviews? Do you have a topic guide you could include alongside the protocol?

As stated in our study aims, on page 7, the purpose of the interviews with patients and carers is to: "Explore the impact of an EPaCCS on the experience of receiving EOL care for patients and their carers."

We have not included an interview topic guide as development of this necessitates an iterative approach, whereby data from the preceding focus group and healthcare professional interviews will inform the development of the topic guide for patients and carers.

Phase 4 – can you clarify in the protocol whether data has already been secured or whether this is to be arranged? Can further detail be provided in the characterisation of EPaCCS and non-EPaCCS patients? How will you determine this?

Agreements are in place with the CCG to obtain the data. Patients will be identified as either having an EPaCCS record (EPaCCS patient) or not (non-EPaCCS patient), using EMIS coding.

Please provide further detail on the integrated digital care record – what is this and how will it enable you to determine who is reviewing EPaCCS records? Why are ambulance HCPs and secondary care included in this list if EPaCCS has not been rolled out to these groups yet? Would you expect them to be accessing EPaCCS?

To provide further detail on the integrated digital care record, we have added the following text to page 7,

“Although some organisations within the CCG area (ambulance service, secondary and social care) are non-EMIS users, information from the EPaCCS template can be viewed across the local health community, via the integrated digital care record used by health and social care professionals in the CCG area, which went live at the end of February 2018. The integrated digital care record contains some of the information held at GP practices, hospital departments, community services, mental health trusts, out of hours services and local authorities across the CCG area, combining it into a single, shared digital record.”

Phase 5 – on what basis have you determined the expected number of deaths (10%) on EPaCCS? This needs clarification if forming the basis of your power calculation.

To explain how we have determined the expected number of deaths as 10% on EPaCCS, we have added the following reference to the text on page 15:

“Of approximately 8,000 deaths occurring in the CCG area over the year of study, we expect around 10% (800 deaths) of patients to have an EPaCCS (10).”

Qualitative methods – will all findings be discussed with PPI representatives to explore and clarify findings – what contributions do you anticipate from the representatives regarding, for example, HCP use (or lack of) by secondary care staff?

Findings will be discussed with PPI representatives to explore and clarify. However, we are unable to comment on what contributions we anticipate the representatives to have regarding the use (or lack of) by secondary care staff.

Dissemination – do you anticipate the value of the research to extend beyond service improvement within the CCG? If so, what pathways to impact do you plan to pursue and how will findings from the study be made relevant for a wide range of audiences?

We expect the refined programme theories to be applicable to EPaCCS and more widely, to include record-sharing and care coordination in general. Learning from the successes and failures of implementing this individual EPaCCS project will enable CCGs to identify what works for whom, why and in which circumstances.

VERSION 2 – REVIEW

REVIEWER	Pablo Millares Martin Whitehall Surgery, Leeds, UK.
REVIEW RETURNED	14-Aug-2019

GENERAL COMMENTS	Some improvement is noticed but I have to wonder, for example, when stating in the abstract "This paper presents a protocol of a mixed-methods study, to understand how, and by whom, EPaCCS are being used" how in the strengths/limitations it is stated that "This study addresses the need for qualitative research into the use of EPaCCS, offering much needed insight into patient and carers' experiences of EPaCCS" but no mention of HCAs; Is this study about clinician's use/benefit or about patients/family/carers perceived impact? I presumed it was both but it is not clearly expressed. There is probably a need to clarify aims section
--

	I would disagree with the statement "Indeed, the need to gather evidence of effectiveness of EPaCCS before widespread and uncritical adoption by the NHS is key" as I consider EPaCCS is already widely disseminated even though evidence about benefit is not so clear in the literature. As it is stated later "83% of Clinical Commissioning Groups (CCGs) in England reported to have an operational EPaCCS, or be in the planning stages, by 2013". Also there is the presumption there is only one EPaCCS, when each area of the country has their own version, their own understanding, their particular interoperability issues to resolve.
--	---

REVIEWER	Matthew Allsop University of Leeds, UK
REVIEW RETURNED	04-Sep-2019

GENERAL COMMENTS	Thank you for your responses to earlier comments. I have raised a few further queries below, grouped by specific section. I have also added some further general comments where further clarification is sought which I think will help with providing a complete description and reflection of planned research activities. Title:  - The title should reflect multiple care sectors within one clinical commissioning group, rather than across England Strengths and limitations:  - First bullet point – it is not clear how the work will inform “EPaCCS commissioning decision-making nationally” and I would consider removing this part of the statement - Fifth statement (about causal pathways) has multiple layers. Could this be simplified to make one point, about small sample sizes or the aims of the study? General comments:  - Thank you for including further detail on the integrated digital care record – please can you confirm whether all data from EPaCCS or only selected items are included. And does the integrated digital care record require additional login processes, which may be a separate process for accessing EPaCCS data as compared to EPaCCS embedded into EMIS or other electronic medical records systems. - How can you be confident that the contexts, mechanisms and outcomes of the EPaCCS you are examining will be applicable to other EPaCCS, when these data have not been obtained yet? - Thank you for your response regarding methods for phase 1. Please can you add detail around the methods used for literature searching into the protocol, for clarity to readers. In terms of the systematic review highlighted, it is a very relevant piece of research related to the study and its incorporation into the
---

	literature overview would ensure it reflects the most up-to-date literature.  - It is not currently reflected in the protocol, but can you add more detail around the time since an EPaCCS was implemented in the CCG. The duration of time since its implementation may affect your findings and relevance to other CCGs thinking of or having implemented EPaCCS. - The quantitative data analysis assumes a home death as one of the outcomes intended from EPaCCS. Without programme theory can you be confident that achieving a home death is the intended outcome of health professionals EPaCCS users or the CCG? - In terms of the PPI involvement in the design of the study, is it possible to add specific detail about which issues were identified and how these informed the design of the study? - In terms of high and low engagement practices, will you be using a denominator to interpret the number of EPaCCS initiated by practice? Further detail in how you will be able to characterise high and low engagement by practices would be helpful. - For data to be obtained from the CCG for EPaCCS, can you confirm that agreements being in place refers to the data being obtained from the CCG or data sharing agreements in place across all CCGs. The latter not being in place may mean you only have access to EPaCCS data for a proportion of practices. - Your study is focusing on one CCG, without comparative analysis across other CCGs or efforts to understand variation in other CCGs. Similar to earlier points linked to the strengths and limitations section and relevance of findings, I would suggest reframing intended impact to reflect tangible local pathways to impact, with potential findings that can inform practice in other CCGs. For example, with the current model of EPaCCS in the CCG you are focusing on, EMIS is used and an interoperability workaround in place (integrated digital care record) for ambulance and secondary care systems. It is not clear how this context differs to other CCGs at the moment and I think it will be difficult to determine the relevance of study findings until comparative work is completed, potentially in the next iteration of this work.
--	--

VERSION 2 – AUTHOR RESPONSE

Reviewer(s)' Comments to Author:

Reviewer: 1

Reviewer Name: Pablo Millares Martin

Institution and Country: Whitehall Surgery, Leeds, UK.

Please state any competing interests or state 'None declared': None declared

Please leave your comments for the authors below

Some improvement is noticed but I have to wonder, for example, when stating in the abstract "This paper presents a protocol of a mixed-methods study, to understand how, and by whom, EPaCCS are

being used" how in the strengths/limitations it is stated that "This study addresses the need for qualitative research into the use of EPaCCS, offering much needed insight into patient and carers' experiences of EPaCCS" but no mention of HCAs; Is this study about clinician's use/benefit or about patients/family/carers perceived impact? I presumed it was both but it is not clearly expressed. There is probably a need to clarify aims section

We are not clear about the issue that the reviewer is raising here. Aim 2 of the study makes clear that both patients/carers and healthcare professionals (HCPs) views and experiences will be sought (page 7):

"Explore the impact of an EPaCCS on the experience of receiving EOL care for patients and their carers, and understand HCPs' views and experiences of utilising an EPaCCS to provide palliative care to their patients."

I would disagree with the statement "Indeed, the need to gather evidence of effectiveness of EPaCCS before widespread and uncritical adoption by the NHS is key" as I consider EPaCCS is already widely disseminated even though evidence about benefit is not so clear in the literature. As it is stated later "83% of Clinical Commissioning Groups (CCGs) in England reported to have an operational EPaCCS, or be in the planning stages, by 2013".

We have amended this statement, which now reads:

"Indeed, the need to gather evidence of effectiveness of EPaCCS is vitally important, as it has already been widely and uncritically adopted by the NHS."

Also there is the presumption there is only one EPaCCS, when each area of the country has their own version, their own understanding, their particular interoperability issues to resolve.

We have made clear in two locations within the paper that this study utilises data from one specific EPaCCS, but that other systems may take other forms:

"An EPaCCS record can take various forms, including web-based electronic registers, systems based on sharing care summaries or care plans, alongside patients' electronic records."

"The study will investigate the impact of only one of the many EPaCCS developed and implemented in the UK."

Reviewer: 3

Reviewer Name: Matthew Allsop

Institution and Country: University of Leeds, UK

Please state any competing interests or state 'None declared': None declared

Please leave your comments for the authors below

Thank you for your responses to earlier comments. I have raised a few further queries below, grouped by specific section. I have also added some further general comments where further clarification is sought which I think will help with providing a complete description and reflection of planned research activities.

Title:

- The title should reflect multiple care sectors within one clinical commissioning group, rather than across England

We have amended the title, so that it now reads:

“THE IMPACT OF ELECTRONIC PALLIATIVE CARE COORDINATION SYSTEMS (EPaCCS) ON CARE AT THE END OF LIFE ACROSS MULTIPLE CARE SECTORS, IN ONE CLINICAL COMMISSIONING GROUP AREA, IN ENGLAND: A REALIST EVALUATION PROTOCOL”

Strengths and limitations:

- First bullet point – it is not clear how the work will inform “EPaCCS commissioning decision-making nationally” and I would consider removing this part of the statement

We disagree with this, as the successes and failures described in one area can be usefully applied to other geographical areas.

- Fifth statement (about causal pathways) has multiple layers. Could this be simplified to make one point, about small sample sizes or the aims of the study?

We have amended this, please see above.

General comments:

- Thank you for including further detail on the integrated digital care record – please can you confirm whether all data from EPaCCS or only selected items are included. And does the integrated digital care record require additional login processes, which may be a separate process for accessing EPaCCS data as compared to EPaCCS embedded into EMIS or other electronic medical records systems.

Given the word limits in place, we have had to be selective about the amount of detail we include about the integrated digital care record. However, we will discuss these issues when we report our findings, if they prove to be relevant.

- How can you be confident that the contexts, mechanisms and outcomes of the EPaCCS you are examining will be applicable to other EPaCCS, when these data have not been obtained yet?

On page 16, we have tentatively suggested the following:

“The synthesised study findings will establish the potential outcomes of EPaCCS, identify the underlying mechanisms which explain how they produce these effects and highlight the key contextual factors that affect their success or failure. Recommendations can then be made for the development and implementation of EPaCCS.”

By clearly describing the context of the EPaCCS under study, we will enable others to make decisions about the relevance of our findings to their circumstances.

- Thank you for your response regarding methods for phase 1. Please can you add detail around the methods used for literature searching into the protocol, for clarity to readers. In terms of the

systematic review highlighted, it is a very relevant piece of research related to the study and its incorporation into the literature overview would ensure it reflects the most up-to-date literature.

We have amended the following text to make clear our methods for the literature search:

“Phase one is complete and included identification of relevant literature from electronic searches of databases, such as Medline and Google Scholar. The search strategy involved searching for papers which discussed or evaluated shared digital records, for the coordination of palliative care, end of life care, or advance care plans. Reference lists of relevant papers were scanned, and citation searches conducted. Grey literature relating to policy and organisational-based material were sought by searching government and other specialist websites.”

We agree that the systematic review is an important addition to the literature, and we will give it prominence when we publish our findings.

- It is not currently reflected in the protocol, but can you add more detail around the time since an EPaCCS was implemented in the CCG. The duration of time since its implementation may affect your findings and relevance to other CCGs thinking of or having implemented EPaCCS.

We agree that this is important information and we will report it when we publish our findings.

- The quantitative data analysis assumes a home death as one of the outcomes intended from EPaCCS. Without programme theory can you be confident that achieving a home death is the intended outcome of health professionals EPaCCS users or the CCG?

We recognise the controversy around place of death as a proxy for quality of care or ‘good death’. However, as it has been noted in the systematic review referred to above, dying at home remains a critical benchmark and allows for comparison with previous studies in this field. We will discuss this in greater detail when we publish the findings of our study.

- In terms of the PPI involvement in the design of the study, is it possible to add specific detail about which issues were identified and how these informed the design of the study?

Within the constraints of the word limit, we feel that our current wording adequately describes the PPI involvement:

“These meetings raised several important issues which have been incorporated into the design of this study. Such issues included allowing patients the choice of whether to have a carer sit alongside them during their interview and which HCPs they felt it was important that the study team spoke to, due to the involvement they had in providing care for patients. The meetings also discussed what terms, wording and questions would be acceptable to patients and carers to read and hear in the study information documents and interviews.”

- In terms of high and low engagement practices, will you be using a denominator to interpret the number of EPaCCS initiated by practice? Further detail in how you will be able to characterise high and low engagement by practices would be helpful.

We have amended the text as follows:

“The research team will purposively sample from a list of practices, to include practices that are high-users of EPaCCS (20 or more records initiated) and low users of EPaCCS (fewer than 20 records initiated) based on data compiled by the CCG which makes clear the number of EPaCCS templates initiated according to practice.”

- For data to be obtained from the CCG for EPaCCS, can you confirm that agreements being in place refers to the data being obtained from the CCG or data sharing agreements in place across all CCGs. The latter not being in place may mean you only have access to EPaCCS data for a proportion of practices.

We are unclear of the relevance of this to the protocol and we are constrained by the word limit.

- Your study is focusing on one CCG, without comparative analysis across other CCGs or efforts to understand variation in other CCGs. Similar to earlier points linked to the strengths and limitations section and relevance of findings, I would suggest reframing intended impact to reflect tangible local pathways to impact, with potential findings that can inform practice in other CCGs. For example, with the current model of EPaCCS in the CCG you are focusing on, EMIS is used and an interoperability workaround in place (integrated digital care record) for ambulance and secondary care systems. It is not clear how this context differs to other CCGs at the moment and I think it will be difficult to determine the relevance of study findings until comparative work is completed, potentially in the next iteration of this work.

We thank the reviewer for his comments and we will bear this in mind when reporting our findings.

VERSION 3 - REVIEW

REVIEWER	Pablo Millares Martin Whitehall Surgery, Leeds, UK
REVIEW RETURNED	13-Nov-2019

GENERAL COMMENTS	It is not clear in page 13 what is the timescale for the creation of the 20 EPaCCS records to consider a practice a high user nor how the sample will be balanced.
--

REVIEWER	Matthew Allsop University of Leeds, UK
REVIEW RETURNED	10-Dec-2019

GENERAL COMMENTS	Many thanks for taking the time to respond to and address previous comments. There are just a few points I would ask for clarification or response to that I did not feel were fully addressed in your most recent response. 1) With regards to the first bullet point of the strength and limitation section you state that findings can be “usefully applied to other geographical areas”. But this is pre-empting findings without a clear rationale. As previously queried, on what basis will you be able to compare or determine if the EPaCCS model in the CCG you are focusing on is similar to any others in other areas of England? You suggest that other areas can make “decisions about the relevance of our findings to their circumstances” – but you can’t assume relevance before allowing others to review your findings. I would suggest removing the reference to “informing EPaCCS commissioning decision-making nationally” or at least revise it to suggest you will explore the relevance of findings to EPaCCS commissioning decision-making nationally during the project.
--

	2) Is it possible to provide further detail of how you categorise high and low engagement practices? This seems to be an important underpinning element of your sampling. Is 20 EPaCCS records an arbitrary number? How was this decided on? And does it take account of practice-specific information that might be important (e.g. patient demographics at a practice or average number of deaths annual). For example, having a much lower number of EPaCCS records at, for example, a student medical practice near a university could still be indicative of high engagement with EPaCCS at a practice. 3) A further point about the title. You have helpfully amended it to reflect that the study is taking place in one CCG but it uses the acronym for EPaCCS which is plural. Could the title be, "THE IMPACT OF AN ELECTRONIC PALLIATIVE CARE COORDINATION SYSTEM ON CARE AT THE END OF LIFE ACROSS MULTIPLE CARE SECTORS, IN ONE CLINICAL COMMISSIONING GROUP AREA, IN ENGLAND: A REALIST EVALUATION PROTOCOL". It may also be worthwhile reviewing the manuscript for how 'EPaCCS' is used – for example in Table 1 where it is referred to as a single entity several times.
--	--

VERSION 3 – AUTHOR RESPONSE

Reviewer 1

It is not clear in page 13 what is the timescale for the creation of the 20 EPaCCS records to consider a practice a high user nor how the sample will be balanced.

We thank the reviewer for his comments. We are not clear on what is meant by the query regarding how the sample will be balanced, but we have tried to make clearer the timescale for the creation of the EPaCCS records, as follows:

“The research team will purposively sample from a list of practices, to include practices that are high-users of EPaCCS and low users of EPaCCS. High EPaCCS use will be defined as practices that have created greater than 20 EPaCCS records (the median number of records across all practices) 4 months post-implementation, based on data extracted by the CCG.”

Reviewer: 3

Many thanks for taking the time to respond to and address previous comments. There are just a few points I would ask for clarification or response to that I did not feel were fully addressed in your most recent response.

1) With regards to the first bullet point of the strength and limitation section you state that findings can be “usefully applied to other geographical areas”. But this is pre-empting findings without a clear rationale. As previously queried, on what basis will you be able to compare or determine if the EPaCCS model in the CCG you are focusing on is similar to any others in other areas of England? You suggest that other areas can make “decisions about the relevance of our findings to their circumstances” – but you can’t assume relevance before allowing others to review your findings. I would suggest removing the reference to “informing EPaCCS commissioning decision-making nationally” or at least revise it to suggest you will explore the relevance of findings to EPaCCS commissioning decision-making nationally during the project.

We have amended the first bullet point of the strength and limitations section so that it now reads:

“Using a theory-driven realist evaluation approach, findings from this study are expected to generate contextually relevant evidence for other care coordination systems.”

2) Is it possible to provide further detail of how you categorise high and low engagement practices? This seems to be an important underpinning element of your sampling. Is 20 EPaCCS records an arbitrary number? How was this decided on? And does it take account of practice-specific information that might be important (e.g. patient demographics at a practice or average number of deaths annual). For example, having a much lower number of EPaCCS records at, for example, a student medical practice near a university could still be indicative of high engagement with EPaCCS at a practice.

To address this matter, we have amended the text on page 14, so that it now reads:

“The research team will purposively sample from a list of practices, to include practices that are high-users of EPaCCS and low users of EPaCCS. High EPaCCS use will be defined as practices that have created greater than 20 EPaCCS records (the median number of records across all practices) 4 months post-implementation, based on data extracted by the CCG.”

3) A further point about the title. You have helpfully amended it to reflect that the study is taking place in one CCG but it uses the acronym for EPaCCS which is plural. Could the title be, “THE IMPACT OF AN ELECTRONIC PALLIATIVE CARE COORDINATION SYSTEM ON CARE AT THE END OF LIFE ACROSS MULTIPLE CARE SECTORS, IN ONE CLINICAL COMMISSIONING GROUP AREA, IN ENGLAND: A REALIST EVALUATION PROTOCOL”. It may also be worthwhile reviewing the manuscript for how ‘EPaCCS’ is used – for example in Table 1 where it is referred to as a single entity several times.

It is our understanding, informed by the wider literature, that EPaCCS is an acronym for both Electronic Palliative Care Coordination System AND Electronic Palliative Care Coordination Systems. We therefore have not changed the title, or how it is used throughout the manuscript.